# ASPiRe:
# Adaptive Skill Priors for Reinforcement Learning

**Mengda Xu** [1, 2]**, Manuela Veloso** [2,3]**, Shuran Song** [1]
[1] Department of Computer Science, Columbia University
[2] J.P. Morgan AI Research [3] School of Computer Science, Carnegie Mellon University (emeritus)
`https://aspire.cs.columbia.edu/`

## Abstract

We introduce ASPiRe (Adaptive Skill Prior for RL), a new approach that leverages prior experience to accelerate reinforcement learning. Unlike existing methods that learn a single skill prior from a large and diverse dataset, our framework learns a library of different distinction skill priors (i.e., behavior priors) from a collection of specialized datasets, and learns how to combine them to solve a new task. This formulation allows the algorithm to acquire a set of specialized skill priors that are more reusable for downstream tasks; however, it also brings up additional challenges of how to effectively combine these unstructured sets of skill priors to form a new prior for new tasks. Specifically, it requires the agent not only to identify which skill prior(s) to use but also how to combine them (either sequentially or concurrently) to form a new prior. To achieve this goal, ASPiRe includes Adaptive Weight Module (AWM) that learns to infer an adaptive weight assignment between different skill priors and uses them to guide policy learning for downstream tasks via weighted Kullback-Leibler divergences. Our experiments demonstrate that ASPiRe can significantly accelerate the learning of new downstream tasks in the presence of multiple priors and show improvement on competitive baselines.

## 1 Introduction

Transferring prior experience to new tasks is central to an agent's adaptability. In this work, we aim to accelerate online reinforcement learning by leveraging prior experience from large offline data. Previous works [1, 2, 3] have proposed to extract temporally extended actions into *skills* from offline datasets and learn a single behavior prior over the skill space, which we refer to as *skill prior*. The skill prior is then used to guide the learning for a new downstream task by constraining the learned policy to stay close to the prior distribution. However, regularizing with a single skill prior might be insufficient for more complex downstream tasks. We conclude two major limitations for using a single skill prior: (1) The datasets could be generated by multiple policies for diverse tasks. As a consequence, the action distribution from behavior trajectories might be multi-modal, and the resulting learned prior might learn an "average" behavior [4] (2) Complex tasks not only require reusing previous skills but also composing them. Consider a task that requires the agent to traverse a maze with obstacles inside. The agent needs to compose navigation skills and avoid skills concurrently when the obstacle is observed (See Fig. 1) and only activate navigation skills when it is not. Guided with only navigation skill prior or avoid skill prior might lead to inefficient learning.

This observation suggests guiding the learning with multiple skill priors, with each prior specialized in one task. During the learning of new downstream tasks, the most relevant priors should be selected to regularize the policy based on the task and specific states. Specifically, it requires the agent to: (1) **Identify which skill prior(s) to use.** Since not all skill priors are relevant to the new task at hand, imposing irrelevant skill priors to regularize can mislead the learning process. (2) **Learn how**

36th Conference on Neural Information Processing Systems (NeurIPS 2022).

**to combine multiple skill priors.** Many complex tasks require the agent to identify and combine multiple priors to solve them effectively. The integration of skill priors can be either **sequential** (active one prior at a time) or **concurrent** (active multiple priors at the same time), which need to be determined by the agent automatically.

To address the above challenges, we propose **ASPiRe**, a new method to accelerate the learning of new downstream tasks by guiding the policy with an adaptive skill prior. To implement this idea, we first extract the multiple primitive skill priors from the separated and labeled offline data. Each primitive skill prior is specialized in solving one specific task (e.g., navigation prior and avoid prior). For a new downstream task, the policy is trained with a new task reward and regularized by the adaptive skill prior. The adaptive skill prior can be one of the primitive skill priors or constructed by composing primitive skill priors.

Our method includes an Adaptive Weight Module (AWM) that infers an adaptive weight assignment over learned primitive skill priors and uses these inferred weights to guide policy learning through weighted Kullback-Leibler (KL) divergence. The weight applied to the KL divergence between the learned policy and a given primitive skill prior determines its impact on the learned policy. This implicitly composes multiple primitive skill priors into a *composite skill prior* and guides the policy learning. Our formulation allows the algorithm to combine the primitive skill priors both sequentially and concurrently as well as adaptively adjust them during the online learning phase.

We evaluate our method in challenging tasks with sparse rewards. We demonstrate that our method can accelerate the learning of new downstream tasks in the presence of multiple priors and show improvement over competitive baselines on both learning efficiency and task success rate.

## 2   Related work

**Offline reinforcement learning.** Offline reinforcement learning [4] exploits an existing offline dataset and learns the best policy from it without interacting with environments. This is valuable when interacting with the environment for a task is expensive. Many algorithms have been developed, both in model-free [5, 6, 7, 8, 9, 10, 11, 12, 13] and model-based [14, 15, 16, 17, 18] settings.

Offline RL has also demonstrated its ability to accelerate online learning [2, 3, 19]. While most of the previous works only consider a behavior prior for a single task, the policy is guided by the behavior prior to achieve efficient exploration. Rao et al. [20] proposes to use a mixture latent variable model to compose multiple behavior priors, but this framework only activates one behavior prior at a time. How to leverage multiple behavior priors during online learning both sequentially and concurrently remains an open question. Our method provides a solution to adaptively select a behavior prior (i.e., skill prior) or compose a new prior to guide the policy learning.

**Reusing and Composing skills.** It is crucial for intelligent agents to reuse and compose learned skills to solve new tasks. A wide range of works have been proposed to solve this longstanding problem [21, 22]. One way to compose and reuse skills is using value functions [23, 24, 25] or cumulants [26, 27, 28, 29]. Numerous works also explore the hierarchical structure in which a high-level policy integrates a collection of low-level controllers (primitives) [30, 31, 32, 33, 34] and each of low-level controllers contains a distinct skill. However, most of the hierarchical methods only sequentially activate the primitives. The other line of work for reusing and composing skills is to operate on a latent space, which can be mapped to action space [35, 36, 37, 38, 39, 40]. Among those works, several approaches employ the idea of mixture-of-experts [41] by explicitly modeling a library of primitive skills and learning a weight gate function to compose the skills [42, 43, 42, 44]. These works show that this additional structure is beneficial for transferring skills to complex tasks. Our work can also be viewed through the mixture-of-experts lens, i.e., solve downstream tasks by composing a library of primitives. However, our work treats the composition as a regularization to guide the policy learning, instead of executing the composition as a policy directly.

## 3   Approach

The goal for ASPiRe is to accelerate downstream task learning using a library of primitive skill priors. There are two main challenges for this task: 1) Decide which primitive skill prior should be activated given a new downstream task, where using unrelated primitive skill prior could mislead the learning

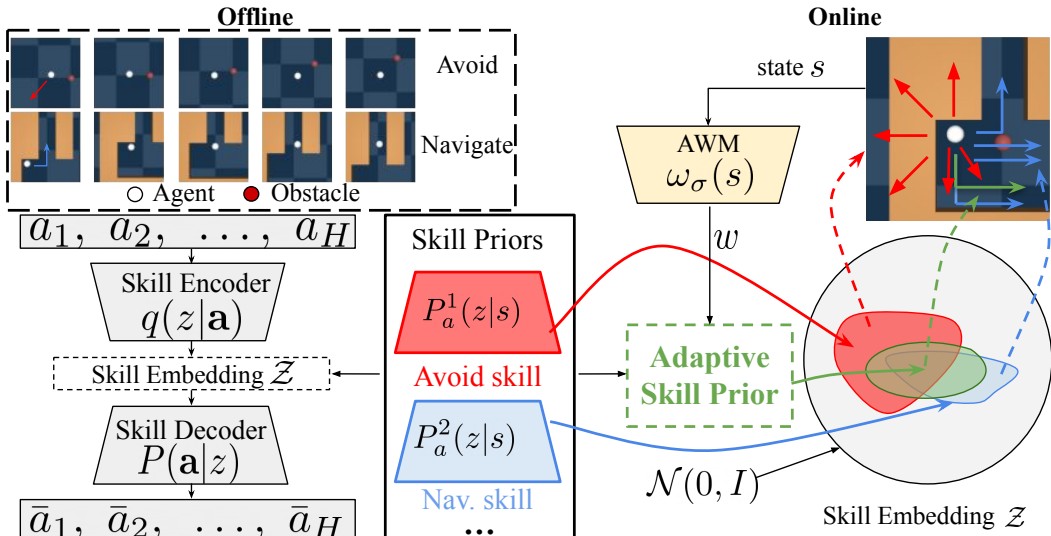

Figure 1: **ASPiRe overview.** In offline, we learn a shared skill embedding space and a collection of distinct primitive skill priors, e.g. , avoid and navigation primitives are learned. The skill encoder embeds an action sequence $\mathbf{a}$, which is sampled from aggregated dataset, into a latent vector in embedding $\mathcal{Z}$ and the action sequence get reconstructed by the skill decoder. The primitive skill prior $P_a^i(z|s)$ imitates the state-skill (in embedding space) pairs in dataset $\mathcal{D}_i$. In online stage, ASPiRe composes multiple primitive skill priors via weighted KL divergence, where weighs are inferred by Adaptive Weight Module (AWM).

process. 2) How to best combine multiple primitive skill priors, either sequentially or concurrently. In particular, to the best of our knowledge, how to simultaneously constrain the policy learning with multiple skill priors is an open challenge.

To address these challenges, we propose a method that regularizes the policy learning using an "adaptive" skill prior, which can be either one of the primitive skill priors or a newly created skill prior by the composition of multiple primitive skill priors (i.e., the composite skill prior).

Our approach consists of two learning stages: offline and online (See Fig. 1). In the offline stage, the system learns multiple primitive skill priors from a set of labeled datasets. Each dataset contains expert demonstrations of one specific primitive task. In the online stage (for downstream tasks), the algorithm creates an adaptive skill prior by assigning weight over primitive skill priors. The weights are generated by the adaptive weight module (AWM). Then they are used to regularize the policy training. In the running example in Fig. 1, the policy is regularized by avoid and navigation primitives **concurrently** when the obstacle is observed.

**Problem Formulation:** In the offline learning stage, the algorithm has access to K datasets $\{\mathcal{D}_1, ..., \mathcal{D}_K\}$ with task labelling $\{\mathcal{T}_0, .., \mathcal{T}_K\}$. Each dataset $\mathcal{D}_i$ contains action-state trajectories $\tau = \{(s_0, a_0), ..., (s_T, a_T)\}$ for task $\mathcal{T}_i$ (i.e. push). The datasets are generated by expert behavior policy $\mu_1, .., \mu_K$. We further assume that none of the single behaviors can individually solve the typically more complex downstream learning tasks.

The online learning stage for the downstream task is formulated as a reinforcement learning problem, modelled as an infinite-horizon Markov Decision Process (MDP) $\mathcal{M} = (\mathcal{S}, \mathcal{A}, \mathcal{P}, \mathcal{R}, \rho, \gamma)$, where $\mathcal{S}$ is the continuous state space, $\mathcal{A}$ is the continuous action space, $\mathcal{P}$ is the dynamic $p(s_{t+1}|s_t, a_t)$, $\mathcal{R}$ is the scalar reward $r_t$, $\rho$ is the initial state distribution and $\gamma$ is the discount factor. The goal is to learn a policy $\pi(a_t|s_t)$ to maximize the expected discounted sum of rewards $\sum_{t=0}^{T} \mathbb{E}_\pi[\gamma^t r(s_t, a_t)]$.

### 3.1 Extracting multiple primitive skill Priors

We denote an action sequence $\{a_t, ..., a_{t+H-1}\}$ with fixed horizon $H$ as a skill $\mathbf{a}_t$. In the offline learning phase, we first learn a shared low-dimensional skill embedding space $\mathcal{Z}$ from aggregated

datasets $\{\mathcal{D}_i\}_{i=0}^K$. We then construct primitive skill priors on top of the skill embedding space for each dataset $\mathcal{D}_i$. Distinct primitive skill priors can generate different types of skills based on the task associated with $\mathcal{D}_i$.

To learn the skill embedding, following the previous works [1, 2, 45], we fit a variational autoencoder and maximize the following evidence lower bound (ELBO):

$$\log p(\mathbf{a}_t) \geq \mathbb{E}_q \Big[ \underbrace{\log p(\mathbf{a}_t|z)}_{\text{reconstruction}} - \beta \big( \underbrace{\log q(z|\mathbf{a}_t) - \log p(z)}_{\text{regularization}} \big) \Big]. \tag{1}$$

Here, $\beta$ is a parameter that controls the regularization term. The skill encoder $q(z|\mathbf{a}_t)$ embeds the skill $\mathbf{a}_t$ into latent space $\mathcal{Z}$ and the skill decoder $p(\mathbf{a}_t|z)$ reconstructs the latent vectors back to skills. We set $p(z)$ as a unit Gaussian prior to regularize the encoder learning. Note that action sequences $\mathbf{a}$ are randomly sampled from all offline dataset $\{\mathcal{D}_i\}_{i=0}^K$ such that diverse action sequences can be embedded in a shared skill embedding space with the same encoder and decoder.

We freeze the skill encoder and decoder parameters after sufficient training. Next, we extract multiple primitive skill priors on top of the shared skill embedding. For each offline dataset $\mathcal{D}_i$, we extract a primitive skill prior $P_a^i(z|s_t)$, which is parameterized as a Gaussian distribution. To train primitives, we sample state and skill tuples $(s_t, \mathbf{a}_t)$ from $\mathcal{D}_i$ and minimize the KL divergence between the skill posterior and the predicted prior $\mathbb{E}_{(s_t, \mathbf{a}_t) \sim \mathcal{D}_i} D_{\text{KL}}\big(q(z|\mathbf{a}_t), p_a^i(z|s_t)\big)$, similar to previous work [2].

## 3.2 Regularized reinforcement learning via adaptive skill prior

For downstream task learning, instead of learning a policy on action space $\mathcal{A}$, we learn a policy $\pi_\theta(z|s_t)$ which acts on high-level skill embedding space $\mathcal{Z}$. Previous work [2] has demonstrated that learning on the skill embedding space is beneficial for complex long-horizon tasks. The skill embedding $z$ can be decoded into an action sequence $\mathbf{a}$ via the skill decoder $p(\mathbf{a}|z)$ for $H$ steps action execution. We further replace the dynamic in our problem formulation with H-step dynamic $p'(s_{t+H}|s_t, z_t)$ and rewards with H-step rewards $r_t' = \sum_{t=1}^H r_t$.

We guide the policy learning via an adaptive skill prior over the course of online learning for downstream tasks. Ideally, the policy should be regularized by all relevant primitive skill priors at each state. We consider two scenarios: 1) only one primitive prior is relevant, 2) multiple primitive priors are relevant. Previous work [2] considers the first scenario and regularizes the learning policy by providing an additional negated KL divergence between the policy and the offline skill prior. Similarly, SVPG [46] regularizes the policy parameters by a prior distribution and optimize the object via SVGD [47].

In states where multiple primitive skills are involved, the agent needs to achieve several goals simultaneously, which are implicitly encoded in distinct primitive skill priors. It is therefore natural to guide the policy learning with multiple primitive skill priors. To implement this idea, we can augment the standard RL objective with a negated *weighted KL divergence*:

$$J(\pi) = \sum_{t=0}^T \mathbb{E}_\pi \Big[ r(s_t, z_t) - \alpha \underbrace{\sum_{i=1}^K \omega_i(s_t) D_{KL}(\pi(z_t|s_t), p_a^i(z_t|s_t))}_{\text{weighted KL divergence}} \Big] \tag{2}$$

Here, we refer to $\omega(s)$ as the *weighting function*, which specifies the weights $w = (w_1, w_2, ..., w_K)$ to apply on KL divergence between the policy and primitive skill priors. We consider weights are bounded $\{w_i\}_{i=1}^K \in [0, 1]$ and are normalized $\sum_{i=1}^K w_i = 1$. Assigning weights to the primitive skill priors implicitly forms a *composite skill prior*. This can be interpreted as the learned policy will be constrained by a skill distribution $p_c(z_t)$ that can minimize the weighted KL divergence given weights $w$ at state $s_t$:

$$p_c(z_t) = \arg\min_{p_c} \sum_{i=1}^K w_i D_{KL}(p_c(z_t), p_a^i(z_t|s_t)) \tag{3}$$

We refer to it as *composite skill prior* generated by weight $w$ at state $s_t$. Weights at a given state can be viewed as a measure of how much one primitive skill prior should contribute to the composite skill prior. The state-dependent weights can be learned simultaneously with the policy, and we discuss it in section 3.3. For now, we assume the weighting function $\omega(s_t)$ is given, and the goal is to maximize the Equation 2.

We follow the policy constraints formulation [5, 6, 10] to optimize the objective. We consider a Q-function $Q_\phi$ and a policy $\pi_\theta$ parameterized by the network with parameters $\phi$ and $\theta$, respectively. The policy parameters can be optimized by minimizing the weighted KL divergence penalty:

$$J_\pi(\theta) = \mathbb{E}_{s_t \sim D}\left[\mathbb{E}_{z_t \sim \pi_\theta}\left[ -Q_\phi(s_t, z_t) + \alpha \sum_{i=1}^{K} \omega_i(s_t) D_{KL}(\pi_\theta(z_t|s_t), p_a^i(z_t|s_t))\right]\right] \tag{4}$$

Here $\alpha \geq 0$ is a temperature parameter to constrain the weighted KL-divergence, which can be tuned automatically by providing a target divergence parameter $\delta$ [2] (See appendix A.3 for details).

Q-function parameters can be optimized by minimizing the Bellman residual:

$$J_Q(\phi) = \mathbb{E}_{s_t, z_t \sim D}\left[\frac{1}{2}\big(Q_\phi(s_t, z_t) - (r(s_t, z_t) + \gamma Q_{\bar\phi}(s_{t+1}, \pi_\theta(z_{t+1}|s_{t+1})))\big)^2\right] \tag{5}$$

### 3.3 Adaptive Weight Module

In this section, we introduce Adaptive Weight Module (AWM), a component to infer the optimal weights adaptively over the course of tasks. Our goal is to learn a parameterized weighting function $\omega_\sigma$ to compose the primitive skill priors via weighted KL divergence, and the resulting composite skill prior can guide the policy learning. We evaluate the expected critic value of skills under the composite skill prior distribution generated by the weights $\omega$. The weights leading to the highest critic value is selected to assign among primitives. Both the weighting function $\omega_\sigma$ and the skill prior generator $G_\eta$ are learned online along with the policy $\pi_\theta$.

**Learning skill prior generator.** To efficiently evaluate the skills under composite skill priors, we implement skill prior generator $G_\eta(z|s_t, w)$, a generative model which outputs the composite skill prior generated by the weights $w$ given the state $s_t$. This can be optimized by minimizing the weighted KL divergence between output distribution $G_\eta(\cdot|s_t, w)$ and primitive skill priors:

$$J_G(\eta) = \mathbb{E}_{s_t \sim \mathcal{D}} \sum_{i=1}^{K} w_i D_{KL}(G_\eta(z|s_t, w), P_a^i(z_t|s_t)) \tag{6}$$

We parameterize the skill prior generator as a Gaussian distribution by the network with parameters $\eta$. During the learning, we randomly sample weights $w$ from the unit simplex as the weight input. This helps the skill prior generator to learn all possible compositions across states.

**Learning adaptive skill prior.** Our goal is to construct an adaptive skill prior that is biased towards the high reward trajectories, similar to Siegel et al. [10]. The key difference is that our approach aims to construct such adaptive prior for a novel downstream task by composing multiple primitive skill priors. In effect, we would like to find weights assignment such that the corresponding composite skill prior can contain the skills that are more likely to lead to task success. We formulate this as:

$$\sigma = \arg\max_\sigma \sum_t \mathbb{E}_{s_t \sim \mathcal{D}}\big[\mathbb{E}_{z \sim G_\eta(\cdot|s_t, \omega_\sigma(s_t))}[Q_\phi(s_t, z)]\big] \tag{7}$$

where $Q_\phi$ is the policy critic defined in section 3.2. The Q-value in the objective measures the usefulness of skills for the task at hand. The inner expectation can be viewed as a measurement of the usefulness of the composite skill prior generated by weight $w$. The objective prefers a composite skill prior, in which skills have higher Q-value and are more promising to be explored. We approximate the inner expectation with $M$ samples from $G_\eta$, i.e, $\mathbb{E}_{z \sim G_\eta(\cdot|s_t, \omega_\sigma(s_t))}[Q_\phi(s_t, z)] = \frac{1}{M}[Q_\phi(s_t, z)|z \sim G_\eta(\cdot|s_t, \omega_\sigma(s_t))]$. We find $M = 20$ is sufficient to provide a stable and accurate estimation.

We parameterize the weighting function $\omega_\sigma$ by the network with parameter $\sigma$, and use softmax as the output layer. To learn the optimal weight, we set the Equation 7 as the loss function. We add an

**Algorithm 1** ASPiRe Algorithm

---

1: **Inputs:** K primitive skill priors $\{P_a^i(z_t|s_t)\}_{i=0}^K$
2: **Initialize:** replay buffer $\mathcal{D}$, policy $\pi_\theta(z_t|s_t)$, critic $Q_\phi(s_t, z_t)$, target critic $Q_{\bar\phi}(s_t, z_t)$, weighting
   function $\omega_\sigma(s_t)$, skill prior generator $G_\eta(z_t|s_t, w_t)$
3: **for** each iteration **do**
4:     **for** every H environment steps **do**
5:         $z_t \sim \pi_\theta(z_t|s_t)$                                                 $\triangleright$ Sample skill from policy
6:         $s'_t \sim p(s_{t+H}|s_t, z_t)$.                                             $\triangleright$ Execute skill
7:         $\mathcal{D} \leftarrow \mathcal{D} \cup \{s_t, z_t, r_t, s'_t\}$                           $\triangleright$ Store experience in replay buffer
8:     **end for**
9:     **for** each gradient step **do**
10:         Sample a batch of transition tuples $\{s_t, z_t, r_t, s'_t\}$ from $\mathcal{D}$
11:         $w_t = \omega_\sigma(s_t)$                                       $\triangleright$ Get weights from weighting function
12:         $\theta \leftarrow \theta - \lambda_\theta \nabla_\theta J_\pi(\theta)$              $\triangleright$ Update policy parameters via weighted KL divergence
13:         $\phi \leftarrow \phi - \lambda_\phi \nabla_\phi J_Q(\phi)$                        $\triangleright$ Update critic parameters
14:         $\eta \leftarrow \eta - \lambda_\eta \nabla_\sigma J(\eta)$                      $\triangleright$ Update generator parameters
15:         $\sigma \leftarrow \sigma - \lambda_\sigma \nabla_\sigma J(\sigma)$                     $\triangleright$ Update weighting parameters
16:         $\bar\phi \leftarrow \tau\phi + (1-\tau)\bar\phi$                          $\triangleright$ Update target critic
17:     **end for**
18: **end for**

---

additional regularization term to penalize weights for deviating from uniform weight $\frac{1}{K}$ in the early stage of the training. This encourages the learned policy to explore more skills in diverse primitive skill priors and prevents weights from collapsing to a singleton due to the noise in Q-value in the early phase. Here, $\beta$ is the annealing coefficient, and it gradually decreases. The weighting function parameters can be trained to minimize the following:

$$J_\omega(\sigma) = \mathbb{E}_{s_t \sim \mathcal{D}} \left[ \mathbb{E}_{z \sim G_\eta(\cdot|s_t, \omega_\sigma(s_t))} \left[ -Q_\phi(s_t, z) + \beta||\omega_\sigma(s_t) - \frac{1}{K}||_2^2 \right] \right] \tag{8}$$

For each gradient step, we optimize both the policy and the AWM. The full Adaptive Skill Prior algorithm is summarized in Algorithm 1. For implementation details, see appendix, A.4.

## 4 Experiments

We would like to answer the following questions through our experiments: a) Can leveraging multiple primitive skill priors accelerate downstream task learning? b) What's the advantage of using the composite skills as regularization compared to directly using them as policy? c) Can AWM optimally assign proper weights over different primitive skill priors based on the downstream tasks, even when irrelevant primitive skill priors are presented?

### 4.1 Experiment setup

We evaluate our method in three modified environments from D4RL [48] and one modified environment from robosuite environment [49]. ASPiRe is given two primitive skill priors in the first two experiments. For the last experiment, we test ASPiRe's ability to compose three primitive skill priors. For each environment, we first collect datasets for distinct skills from expert demonstrations, which allows us to extract distinct primitive skill priors. We then evaluate our method with extracted primitives in downstream tasks.

**Point Maze.** We modified the Point Maze environment by adding multiple obstacles in the maze. The task is to navigate a point mass agent in the maze and reach the goal. During this process, the agent must avoid the obstacles randomly placed in the maze. The original point maze environment is already challenging because of the sparse reward feedback. Randomly placing obstacles in the maze makes this problem even more difficult. For each episode, the maze layout is fixed. However, the agent start, goal, and obstacles locations are randomly specified. Our method will be equipped with two primitive skill priors for this task: navigation and avoid.

**Ant Push.** The ant needs to push the box to the goal while avoiding an obstacle in this task. ASPiRe will carry two primitive skill priors: push and avoid. The ant, box, obstacle, and goal locations are randomly generated at the beginning of every episode. The agent will receive a sparse reward upon the box reaching the goal location.

**Ant Maze.** We modified the Ant Maze by adding an obstacle and a box in the maze. The agent's goal is to traverse the maze while pushing the box to the goal position without hitting the obstacle. The agent always starts in the lower right corner of the maze. However, the box, goal, and obstacle positions are randomly selected from a pre-defined set. Ant Maze itself poses a complex exploration problem due to sparse reward feedback. We make this problem even more complex by changing the success criteria. In this case, ASPiRe will carry three primitives: push, navigation, and avoid.

**Robotic Manipulation.** We further test ASPiRe's ability in complex robotic manipulation task in robosuite environment [49]. The task is to control a Panda robot arm to grasp the box (in red) without colliding with the barrier (in green) placed in the middle of the desk (See sub-figure 1 in Fig. 5).The location of the box and barrier are randomly generated at the beginning of the episode. ASPiRe will carry two primitive skill priors: grasp and avoid.

See appendix A.4 for details on environment, data collection process and training.

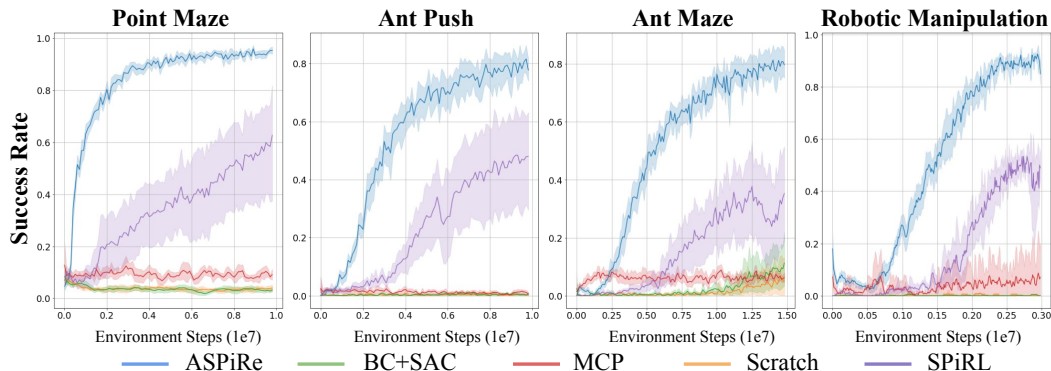

Figure 2: **Comparative Experiment.** Learning curves of our method and comparative methods for downstream tasks. ASPiRe significantly improves the learning efficiency and performance on all challenging tasks. The shaded area is the standard deviation over eight seeds.

**Alternative approaches.** We compare ASPiRe with several prior works for downstream task learning performance. To fairly evaluate the performance, we let all methods (besides SPiRL, as the method learns its own skill embedding) act on skill embedding space $\mathcal{Z}$ learned by the ASPiRe.

- **Scratch.** Train a Soft Actor-Critic (SAC, [50]) agent from scratch. This comparison tests the benefit of using the learned skill priors to guide downstream task learning.

- **Behavioral Cloning with finetuning.** A behavior cloning policy is first trained with offline data and then finetuned using SAC for the downstream tasks.

- **MCP**[51]**.** The method composes primitive policies through multiplicative composition of Gaussians with a learnable weighting function. MCP then samples the actions from the resulting multiplicative composite distribution. For a fair evaluation, MCP is directly given the same set of primitives that ASPiRe is equipped with. This comparison tests the benefit of treating composite skills as prior, instead of directly executing them.

- **SPiRL** [2]**.** Guide the policy learning with a single skill prior. We train the skill prior by combining all primitive datasets into one and learning a single prior from it. This comparison tests the benefit of guiding the learning policy with multiple primitive skill priors.

### 4.2 Experiment results

As shown in Fig. 2, our experiments demonstrate that ASPiRe is able to learn efficiently and converge to high success rates in all challenging environments.

**Benefits of multiple primitive skill priors.** We first compare ASPiRe with methods without skill prior guiding, namely training from scratch and behavior cloning with finetuning. They fail to solve all three tasks (Fig. 2). This confirms the finding in Pertsch et al. [2] that skill prior is beneficial for learning complex and long-horizon tasks. Next, we compare our method with SPiRL [2], which regularizes the learned policy with a single skill prior extracted from the aggregated dataset. By leveraging the skill prior, it can achieve more efficient exploration but still relatively low learning speed and final success rates compared to ASPiRe (Fig. 2). We hypothesize that the performance gap between ASPiRe and SPiRL comes from the prior they deploy to guide the policy learning. We find that SPiRL learns an "average" behavior over all primitives. It prevents SPiRL from guiding states where the specific primitive skills are required. In the case of ASPiRe, it can construct an adaptive skill prior from all primitive skills priors. Therefore, the learned policy can receive sufficient guidance to explore and learn the tasks.

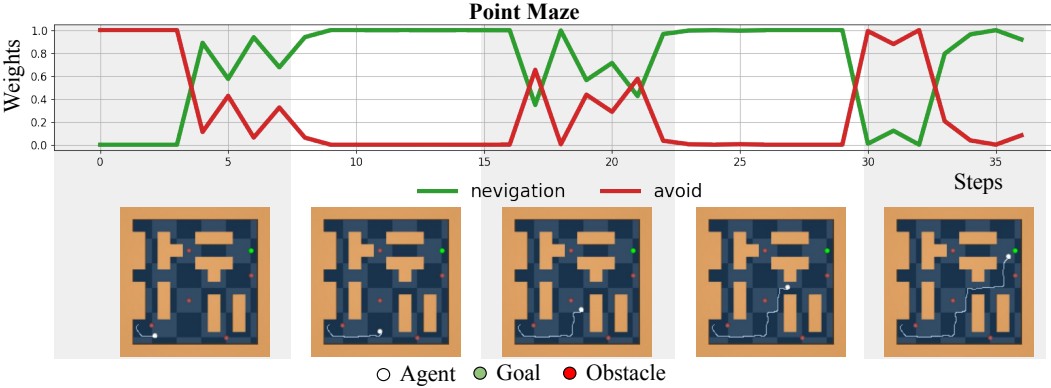

Figure 3: **Weights inference over one episode (Point Maze)**. ASPiRe is able to compose the navigation prior and avoid prior concurrently (grey phases) when the obstacle is observed and only activate navigation prior when it is not (white phases).

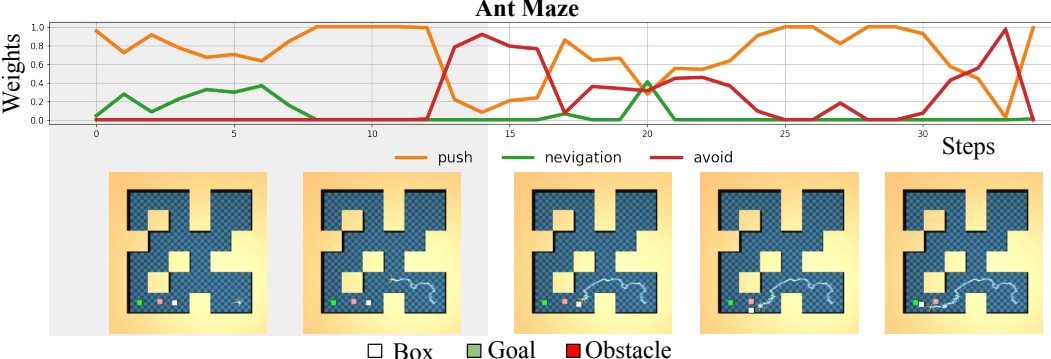

Figure 4: **Weights inference over one episode (Ant Maze)** In the grey phase, ASPiRe is able to compose the navigation prior and push prior concurrently to traverse the maze and reach the box. In the white phase, the model composes the push and avoid prior to push the box to the goal while avoiding the obstacle along the way.

**Benefits of using composite skills as regularization.** We compare ASPiRe with MCP, which directly executes the composite skills as a policy with a learnable weight. As shown in Figure 2, MCP struggles to learn a meaningful policy for all three tasks. This finding demonstrates that using composite skills as regularization provides the necessary flexibility for ASPiRe to learn the policies that can specifically solve the downstream task. We find the final learned policy has lower entropy than the skill priors in most cases. In fact, skill priors provide guidance on the valid skills to explore the environment, but the learned policy needs to further search in this valid skill space to solve the specific problem. We hypothesize that the resulting policy from MCP might be overly broad as the multiplicative composition of Gaussian inherent the relative high entropy from the primitives.

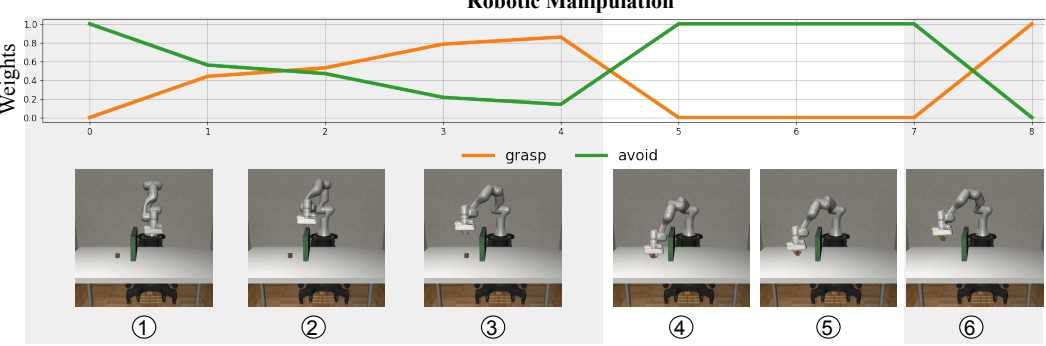

Figure 5: **Weights inference over one episode (Robotic Manipulation)** In the first grey phase (sub-figure 1, 2, 3), ASPiRe is able to compose the grasp prior and avoid prior concurrently to lift the robot arm to avoid the barrier while approach to the box. In the white phase, notice that the robot arm is initially very close to the barrier (sub-figure 4). To avoid any potential collision between the robot arm and the barrier, ASPiRe activates the avoid prior to move the robot arm away from the barrier (sub-figure5). In the end, ASPiRe activates the grasp prior to grasp the box.

**Learning optimal weights.** To demonstrate AS-PiRe's ability to learn optimal weights, we visualize the weights generated by the weighting function for the primitive skills priors at different states for point maze, ant maze and robotic manipulation environments. From Fig. 3, 4 and 5, we find that ASPiRe can compose the primitive skill priors both sequentially and concurrently throughout the tasks. More importantly, ASPiRe regularizes the learned policy with relevant priors, e.g., activates the avoid primitive when obstacles are near the agent. See appendix for more details.

**Training with irrelevant primitive.** To test AS-PiRe's ability in handling irrelevant primitive skill prior, we modify the task of the Ant Maze environment, where the ant needs to reach the goal while avoiding the obstacle along the way without pushing the box. We place the box in the lower right corner of the maze, which is not on the path to the goal. In this case, the push primitive is irrelevant, and regularizing with it will yield low learning efficiency. From Fig. 6, we find that ASPiRe still achieves good performance even with irrelevant primitive. We plot average weights over the states during the training. ASPiRe weights the navigation primitive heavily and assigns a low weight for the irrelevant primitive – push. The weight for avoid is also relatively low since there are fewer states where avoiding is needed (i.e., only when the obstacle is in view).

**Guiding with composite or single primitive.** In this experiment, we test the advantage of using composite skill prior to guiding policy learning compared to

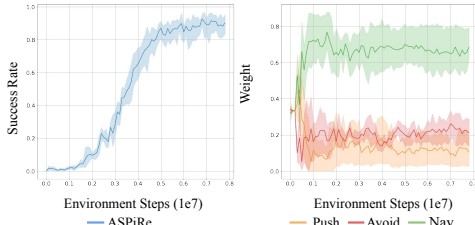

Figure 6: **Training with unrelated primitive.** Left: Learning curve. Right: Average weights of skill priors, where we can observe that ASPiRe assigns a lower weight for unrelated skill prior (e.g., push).

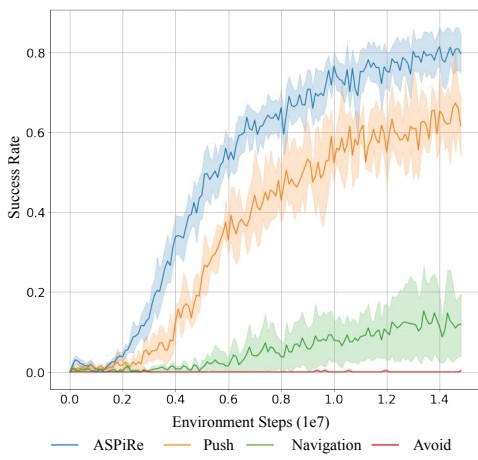

Figure 7: **Guiding with composite or single primitive.** Learning curves in Ant Maze.

guiding with a single primitive skill prior. To answer the question, we guide the policy learning with one of the primitive skill priors (i.e., push, navigation and avoid respectively) in ant maze environment. From Fig. 7, we find that the learned policy guided with only the avoid primitive or navigation primitive fails to learn the task. Guided with the push primitive achieves relatively good performance but fails to complete the setting requiring traverse in the maze. This experiment suggests that composing multiple skill priors is beneficial for the complex multi-task downstream tasks.

**Limitation and future work.** When extracting primitive skill priors, our algorithm requires separated and labeled datasets, each containing a specialized skill. However, such dataset may not always be easy to generate, and manually separating or defining different primitive skill priors can be challenging and sometimes ambiguous. Therefore, an interesting future direction will be allowing the system to process unlabeled dataset and automatically extract a set of distinct primitive skill priors.

## 5   Conclusion

We present ASPiRe, an approach to transfer a spectrum of skills from offline data to accelerate the learning of unseen downstream tasks. We propose to guide the policy learning with multiple primitive skill priors and transfer the diverse skills to RL via weighed KL divergence. We model an Adaptive Weight Module to learn optimal weights assignment and construct an adaptive skill prior. We evaluate ASPiRe in challenging, sparse reward environments and demonstrate the benefits of guiding the learning with multiple skill priors. We also show that ASPiRe is able to select the most relevant skill priors even when unrelated skill priors are presented.

## Acknowledgement

The Authors would like to thank Nelson Vadori, Sumitra Ganesh, Huy Ha, Cheng Chi, Samir Gadre, Zhenjia Xu, and Zhanpeng He on valuable feedback and support. Mengda Xu's work is supported by JPMorgan Chase & Co. This paper was prepared for information purposes in part by the Artificial Intelligence Research group of JPMorgan Chase & Co and its affiliates ("JP Morgan"), and is not a product of the Research Department of JP Morgan. JP Morgan makes no representation and warranty whatsoever and disclaims all liability, for the completeness, accuracy or reliability of the information contained herein. This document is not intended as investment research or investment advice, or a recommendation, offer or solicitation for the purchase or sale of any security, financial instrument, financial product or service, or to be used in any way for evaluating the merits of participating in any transaction, and shall not constitute a solicitation under any jurisdiction or to any person, if such solicitation under such jurisdiction or to such person would be unlawful.

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
