# ASPiRe:
# Adaptive Skill Priors for Reinforcement Learning

**Mengda Xu** [1,2], **Manuela Veloso** [2,3], **Shuran Song** [1]
[1] Department of Computer Science, Columbia University
[2] J.P. Morgan AI Research [3] School of Computer Science, Carnegie Mellon University (emeritus)
https://aspire.cs.columbia.edu/

## 1 Appendix

### 1.1 Hyperparameter Sensitivity Analysis

There are two hyperparameters that can potentially impact the online learning for the downstream task: a) The target KL divergence $\delta$, which is used to automatically tune the temperature parameters in Eq. 4 and b) The sample size $M$ which is used to estimate the critic value in Eq. 7. We investigate the impact of the sample size $M$ on both Point Maze and Ant Maze environment and the choice of the target KL divergence $\delta$.

We first investigate the impact of sample size M by setting $M = 1, 10, 20$. As shown in Fig. 1, We find that the sample size has almost no impact on the learning. We hypothesize two reasons: a) The large train batch size in our experiment helps to estimate the critic value $Q(s, \omega)$ more stable and b) Given the input size of weight vector $\omega$ is small, it might be easy for the neural network to approximate its value via small sample size.

We further investigate the impact of the target KL divergence $\delta$. As shown in Fig. 2, we find that target KL divergence significantly impacts the learning. Based on our experiments, we conclude that a) More complex tasks require higher target KL divergence. Notice that the target KL divergence imposes on Ant Maze is higher than the one on Point Maze. The optimal policy to solve complex tasks might be significantly different from the composite skill prior. Therefore, the policy needs more "space" to explore around the composite skill prior. b) Imposing too small target KL divergence can lead to downgraded performance. From Fig. 2, the ASPiRe's performance drops significantly when we set $\delta = 6$ in Point Maze and $\delta = 10$ in Ant Maze. This is because the policy will be forced to stay close to the composite skill prior and itself might not be able to solve such complex tasks. c) Imposing too big target KL divergence can lead to downgraded performance. We observe that the learning is not efficient when setting $\delta = 32$ in Point Maze and $\delta = 50$ in Ant maze. As target KL divergence increases, the learned policy will receive less guidance from the prior. Though the learning is sensitive to the choice of target KL divergence, we find that there might still be a range of KL divergences values leading to the same optimal performance. Notice the ASPiRe achieves the same good performance when setting $\delta = 12$ or $\delta = 18$ in the Point Maze. We can also observe the same behavior on Ant Maze.

### 1.2 Composite Skill Prior as Policy

We investigate the importance of using composite skill prior as regularization by comparing the ASPiRe with directly executing composite skill prior as a policy. This additional baseline is similar to the MCP. Both of them try to directly sample the action from composite distribution, but the difference is how they composite the distribution. One uses multiplicative gaussian (MCP) the other uses weighted KL divergence(ours).

36th Conference on Neural Information Processing Systems (NeurIPS 2022).

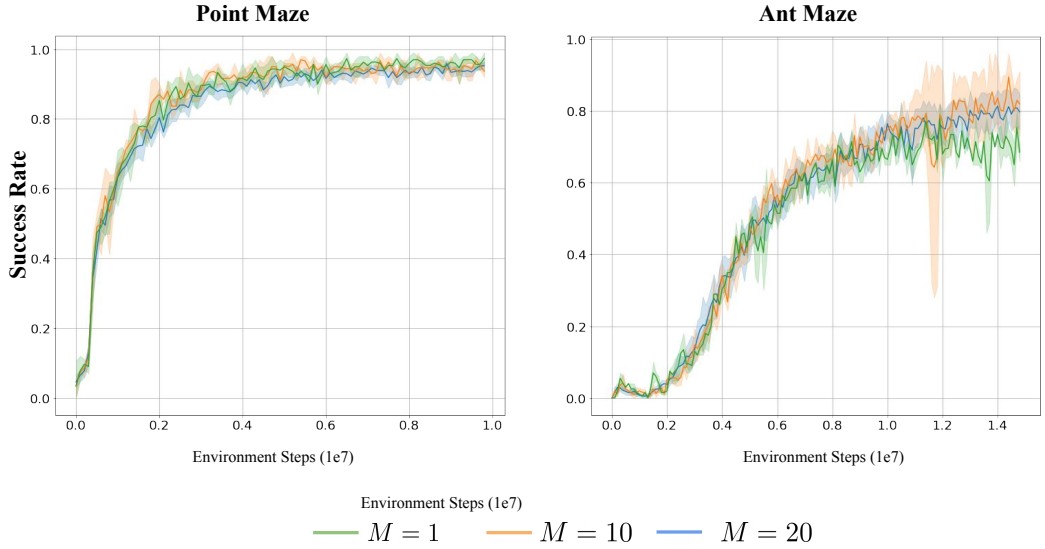

Figure 1: **Sensitivity towards sample size.** Learning curves of our method with different sample size $M$ in Point Maze and Ant Maze. The algorithm is not sensitive to this parameter.

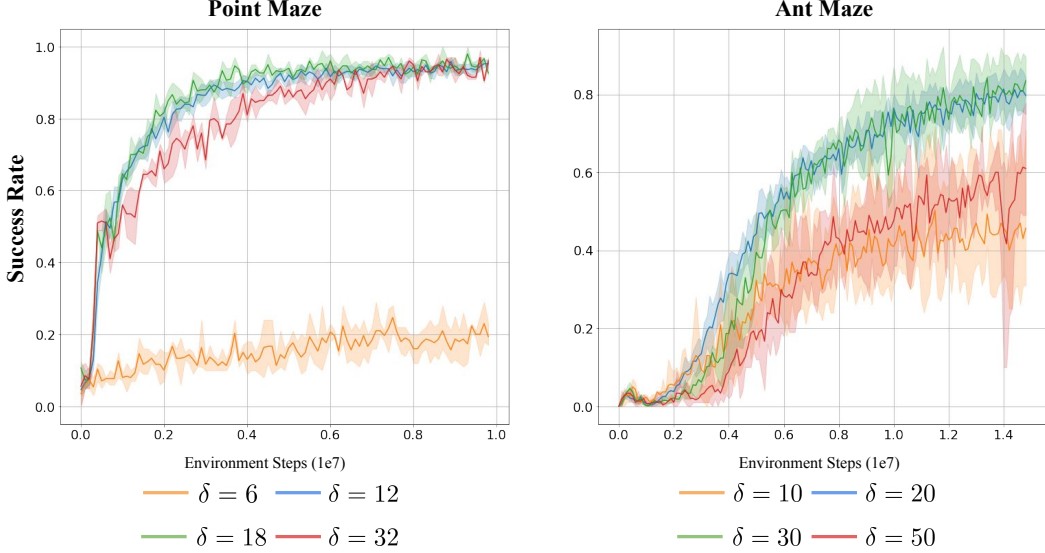

Figure 2: **Sensitivity towards target KL divergence.** Learning curves of our method with different target KL divergence $\delta$ in Point Maze and Ant Maze. More discussion in Sec 1.1.

From Fig. 3, we observe that using composite skill prior as a policy result in poor performance (similar performance as MCP). This confirms the importance of using composite skill prior as prior instead of policy.

## 1.3 Uniform Weights

We have tested the uniform weight in the Ant Maze domain. Specifically, we conduct the experiment in two different settings: a) all skill priors primitives are relevant to the downstream tasks b) not all skill priors are relevant to the downstream tasks.

In setting a) when all primitives are relevant, We observe that uniform weights also deliver good performance. If we compare it with many other random weights assignment (green line in Fig 4),

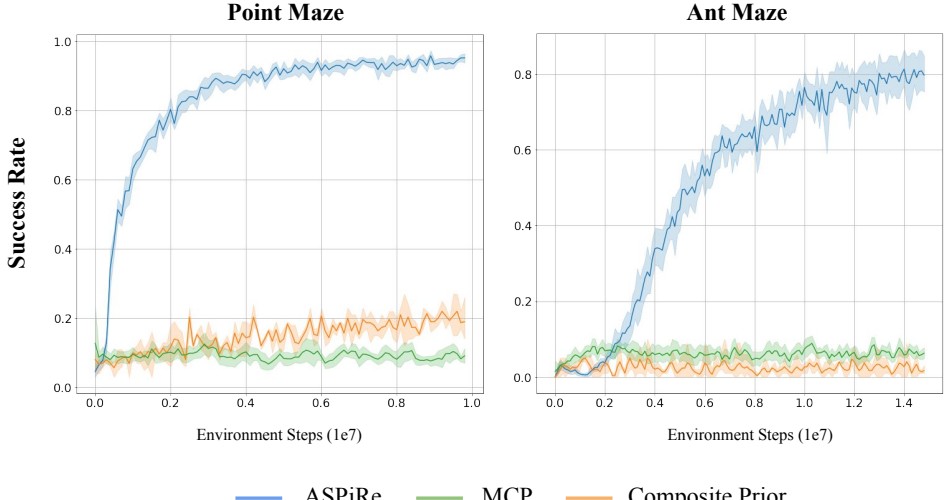

Figure 3: **Composite Skill Prior as Policy.** Learning curves of ASPiRe, MCP and Composite Skill Prior as policy in Point Maze and Ant Maze. Composite skill prior achieves similar performance as MCP which is much worse than ASPiRe.

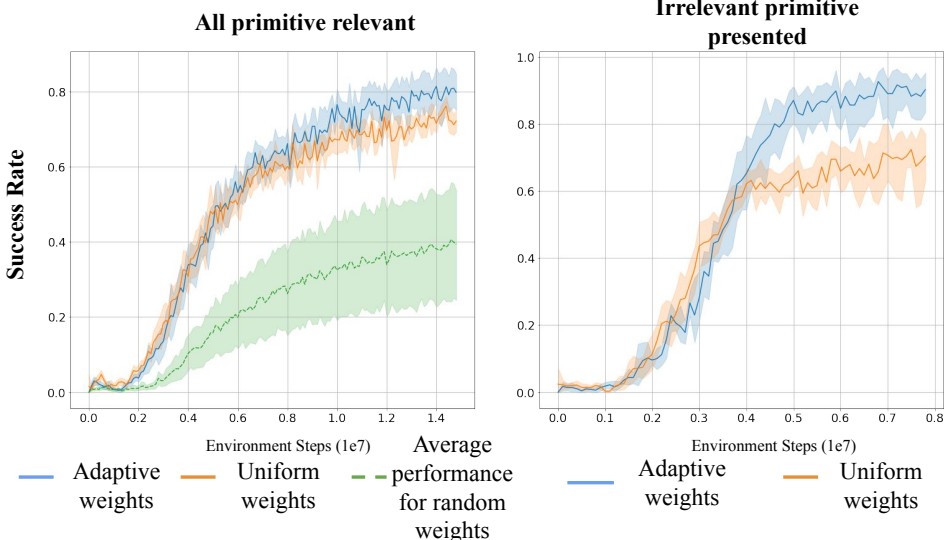

Figure 4: **Adaptive Weights vs Uniform Weights.** Learning curves of our method with Adaptive Weights and Uniform Weights in the setting (a) and (b). Notice that the tasks are different for this two settings. In the setting (a), The agent's goal is to traverse the maze while pushing the box to the goal position without hitting the obstacle. In the setting (b), the agent needs to reach the goal while avoiding the obstacle along the way without pushing the box.

uniform weights achieve a near-optimal assignment. The random weight performance is average of 6 different weight assignments. Regardless, our algorithm can still achieve the best performance among all weights assignments, showing that the system has enough flexibility to learn optimal policy.

In the setting b) not all skill priors are relevant to the downstream tasks. Our experiment shows that the adaptive weights assignment significantly outperforms the uniform weights. This suggests that our adaptive weight assignment is robust to the various downstream tasks.

## 1.4 Implementation Details

### 1.4.1 Offline phase

The skill encoder is a one-layer LSTM with 128 hidden units, which receives $H = 10$ steps action sequence as input and outputs the parameters of the Gaussian posterior $\mathcal{N}(\mu_z, \sigma_z)$ on a 10 dimensional skill embedding. The skill decoder is also a one-layer LSTM with 128 hidden units,

which reconstructs the latent embedding back to the H steps action sequence, i.e., skill. Each primitive skill prior is parameterized with a 6-layer MLP with 128 hidden units per layer. We use ReLU as our activation layer and apply batch normalization. The network outputs parameters of the Gaussian skill prior $\mathcal{N}(\mu_a, \sigma_a)$.

The models are optimized with Adam with learning rate $1 \times 10^{-4}$ and batch size 64. In the offline learning phase, we first learn a shared low-dimensional skill embedding space $\mathcal{Z}$ from aggregated datasets $\{\mathcal{D}_i\}_{i=0}^{K}$. For each iteration, we select one of the primitive datasets $\mathcal{D}_i$ and sample the state-skill tuples from it to optimize the skill embedding and the corresponding primitive skill prior $p_a^i(z|s)$. Once the ELBO loss in section 3.2 converges, we freeze the skill encoder/decoder parameters and only update the parameters of primitive skill prior. This makes sure that primitive skill priors can be learned on a stationary skill embedding space. We set the regularization parameter $\beta = 1 \times 10^{-4}$ for all experiment domains.

### 1.4.2 Online phase

The learned policy $\pi_\theta(s)$ is parameterized with a 6-layer MLP with 128 hidden units per layer. The network outputs the parameters of a Gaussian distribution. We limit the action range of the policy between $[-2, ..., 2]$ by a tanh function. The weighting function $\omega_\sigma(s)$ is also parameterized with a 6-layer MLP with k-way softmax as the output layer. We use ReLU as our activation layer and apply batch normalization for both the learned policy and the weighting function. We model two critic networks and take the minimum value as Q-value estimation, which stabilizes the training. Each critic network is implemented as 4-layer MLP with 256 hidden units per layer. Skill prior generator $G_\sigma(z|s)$ is parameterized with a 3-layer MLP with 128 hidden units per layer. The network outputs the parameters of a Gaussian distribution.

We optimized the models with Adam and set the learning rate as $1 \times 10^{-4}$. The replay buffer capacity is 1e6, and the batch size is 256. We empirically find that setting the discount factor $\gamma$ as 0.97 can slightly stabilize the learning. The annealing coefficient $\beta$ starts with 1 and gradually decreases to $1 \times 10^{-3}$. We set the target divergence $\delta = 12$ for Point Maze, $\delta = 15$ for Ant Push, and $\delta = 20$ for Ant Maze. The temperature parameter $\alpha$ can be tuned automatically by minimizing the following:

$$\arg \min_{\alpha > 0} \mathbb{E}\left[\alpha\delta - \alpha \sum_{i=1}^{K} \omega_i(s_t) D_{KL}(\pi(z_t|s_t), p_a^i(z_t|s_t))\right] \tag{1}$$

The prove can be found in Pertsch et al. [1]. We replace the single term KL divergence in the original formulation with weighted KL divergence. All models are trained on a single NVIDIA GPU.

### 1.5 Environments and data collection

For downstream tasks learning (online phase), the state $s_t$ can be partitioned into two components:

$$s_t = s_t^{\mathrm{p}} + g_t$$

where $s_t^{\mathrm{p}}$ is the proprioceptive observation and $g_t$ is the downstream task-related observation. The states in each primitive dataset $\mathcal{D}_i$ always include the proprioceptive observation $s_t^{\mathrm{p}}$, but part of the downstream task-related observation might be absent. For example, during the learning of Point Maze, the proprioceptive observation $s_t^{\mathrm{p}}$ includes the agent's local view and velocity, and the task-related observation $g_t$ includes the positions of the obstacle, the goal, and the agent. However, in the navigation dataset, the whole task-related observation is absent. Suppose we extract the navigation primitive skill prior based on the proprioceptive observation only. In that case, the resulting prior cannot be directly applied to the downstream task as the state dimensions are different. To remedy this issue, we augment the states in offline data with random vectors that have the same length with the absent task-related observation. This augmentation will not impact the prior learning as the network will eventually ignore random vectors during the extracting process. Alternatively, instead of augmentation, we can parse the states during downstream task learning and only feed the relevant states to primitive skill priors. We have experienced both ways and find the choice does not impact the performance. For the experiment result in this paper, we apply augmentation. This issue might be solved by a method like Attention. However, this is out of the scope of this paper, and we will leave it for future research.

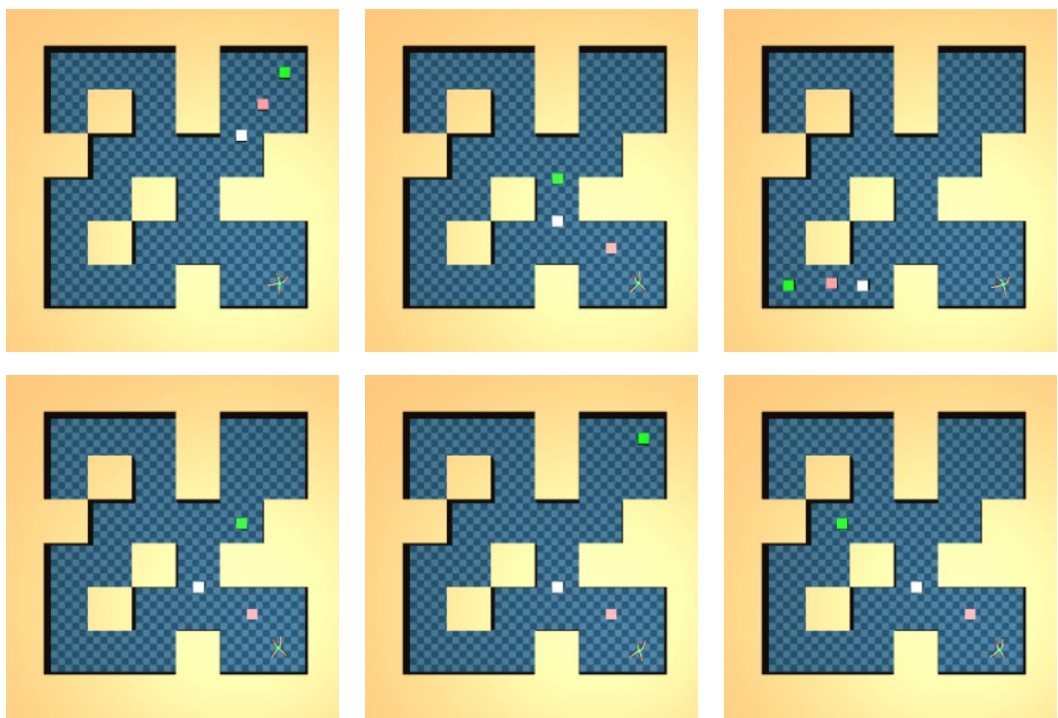

Figure 5: **Pre-defined sub-tasks in Ant Maze**

### 1.5.1 Point Maze

The navigation dataset is collected by running the planning algorithm (provided in D4RL benchmark) in multiple random generated maze layouts. The avoid dataset is collected by a simple heuristic algorithm, i.e., the agent heads in any direction that can prevent itself from colliding with the obstacle. The heading direction $\theta$ can be calculated as: $\theta = -\frac{x_t^{agent} - x_t^{obstacle}}{||x_t^{agent} - x_t^{obstacle}||} \pm \gamma$, where $\gamma$ is a random number sampled from $[-\frac{\pi}{4}, \frac{\pi}{4}]$, $x_t^{agent}$ is the position of the agent, and $x_t^{obstacle}$ is the position of the obstacle. The proprioceptive observation $s_t^{\mathrm{p}}$ in primitive datasets includes the agent's local view and $v_t^{agent}$, which is the velocity of the agent.

For the downstream task, the obstacles, goal, and agent positions are randomly reset at the beginning of one episode. We set a minimum distance between the goal and the agent position as 5 to avoid trivial tasks to constrain the reset process. The point agent will receive a sparse positive reward when the agent reaches the goal and a sparse negative reward when the agent hits the obstacle. The episode will be immediately reset once the agent receives either a positive or a negative reward. The additional task-related state $g_t$ is represented as a 30-dimensional vector which includes the local observation (i.e., goal/obstacle if in view) and the goal position. The action is a 2-dimensional continuous space that controls the agent's velocity in x and y-direction.

### 1.5.2 Ant Push

To collect the push and avoid dataset, we first train a reaching policy by SAC, which allows the ant to head towards a specific target. The reaching policy takes a vector input $(x_t^{goal} - x_t^{agent}, s_t^{\mathrm{p}})$, where $x_t^{goal}$ is the goal position, $x_t^{agent}$ is the agent position, and proprioceptive observation $s_t^{\mathrm{p}}$ is a 29-dimensional vector that represents the ant's joint state. The policy outputs an 8-dimensional vector to control the ant's joint. To collect the push dataset, we randomly place the ant at a distance of $[2, 8]$ from the box. We use the reaching policy to create push behavior by setting the goal as the center of the box. The avoid dataset is generated by the same heuristic policy in the Point Maze environment.

For the downstream task, the obstacle, goal, and agent positions are randomly reset at the beginning of one episode. We randomly initialize the goal position $x_0^{goal}$ at a distance of $[-4, 4]$ from the center $(0, 0)$ and the agent position $x_0^{agent}$ at a distance of $[-8, 8]$ from the center $(0, 0)$. The obstacle is initialized between the goal and the agent $x_0^{obstacle} = \frac{x_0^{goal} + x_0^{agent}}{2} + \epsilon$, where $\epsilon$ is sampled from Uniform(-1,1). Like Point Maze, we set a minimum initial distance between the goal and the agent position as 2 to avoid trivial cases. The agent will receive a sparse positive reward when the agent reaches the goal and a sparse negative reward when the agent hits the obstacle. The additional task-related state $g_t$ is a 4-dimensional vector representing the position difference between the agent and the goal $x_t^{goal} - x_t^{agent}$, and the position difference between the agent and the obstacle $x_t^{obstacle} - x_t^{agent}$. The action is an 8-dimensional continuous space.

### 1.5.3 Ant Maze

We reuse the push and avoid dataset in Ant Push to extract the push and avoid primitive. Navigation primitive is extracted from the dataset (antmaze-medium-diverse) in D4RL benchmark. The proprioceptive observation $s_t^{\mathrm{p}}$ is a 29-dimensional vector that represents the ant's joint state.

For the downstream task learning, we modify the Ant Maze environment from D4RL benchmark by adding an obstacle and a box in the maze. There are 6 pre-defined sub-tasks in the Ant Maze environment, and each of the sub-tasks specifies distinct obstacle, box, and goal positions. The agent will be given a task that is randomly selected from pre-defined sub-tasks at the beginning of every episode. The reward setting is the same as the one in Ant Push. However, the additional maze structure posts a hard exploration problem. The additional task-related state $g_t$ is a 6-dimensional vector representing the position difference between the agent and the goal $x_t^{goal} - x_t^{agent}$, and the position difference between the agent and the obstacle $x_t^{obstacle} - x_t^{agent}$ and the agent position $x_t^{agent}$. The action is an 8-dimensional continuous space.

### 1.5.4 Robotic Manipulation

The task is to control a Panda robot arm to grasp the box without colliding with the barrier placed in the middle of the desk. The height of the barrier is set to be higher than the initial position of the robot arm's end effector. The location of the box and barrier are randomly generated at the beginning of the episode. The agent will receive a sparse positive reward once it grasps the box and lift it up without colliding with the barrier. ASPiRe will carry two primitive skill priors: grasp and avoid. The grasp data is generated by a hand-coded heuristic policy, which first moves the end effector horizontally to the top of the object and then moves the end effector down to grasp the object. The avoid data is also generated by a hand-coded heuristic policy, which moves the end effector up vertically once the barrier is observed. As the datasets are collected by heuristic policies, a part of trajectories might fail to complete the primitive tasks, i.e., grasp the box. Therefore, we only keep the trajectories that successfully complete the primitive tasks.

## References

[1] Karl Pertsch, Youngwoon Lee, and Joseph J. Lim. Accelerating reinforcement learning with learned skill priors. In *Conference on Robot Learning (CoRL)*, 2020.