# OpenReview forum: "ASPiRe: Adaptive Skill Priors for  Reinforcement Learning"
_NeurIPS.cc/2022/Conference — NeurIPS 2022 Accept_

### Official Review · Reviewer_iPXJ · 2022-07-10

**Rating:** 5
**Confidence:** 4
**Soundness:** 3 good
**Presentation:** 3 good
**Contribution:** 2 fair

**Summary:**

The authors present a method for 1) extracting skills from offline demonstration datasets and 2) using these learned skills for accelerated learning of downstream tasks, where the new task might often require composing two or more skills (either concurrently, or sequentially). For (1), the authors first train a VAE on action sequences from the offline dataset to learn a latent space of skills. They then extract primitive skills priors P^i_a(z|s_t) (one for each dataset D_i) by minimizing a KL divergence term (line 140 in the paper). Finally, when learning a new task, the policy that is being learned is constrained to stay close to a mixture of skill priors through a KL divergence term. The weights of this mixture (called the adaptive weight module) is learned so as to maximize the Q-values of the downstream task.

The authors evaluate their method on three simulated navigation tasks. In two of the tasks, two skill priors need to be combined for learning a new task, while there are three skill priors in one of the tasks. The authors provide comparisons against SPiRL (the method on which they closely build), learning from scratch, and naively combining BC with RL, and show that their method significantly outperforms these other techniques.

**Questions:**

- Since I have concerns regarding the overall training stability of the method (due to the large number of moving parts), can the authors share in detail what kind of hyperparameter sweeps had to be done? Generally speaking, a hyperparameter sensitivity analysis would be very insightful for this work.
- Did you try running the method on domains other than navigation? What did you find? Even negative results can be useful to look at here.
- As mentioned in the introduction, one of the relevant related work is PARROT, but there is no comparison provided to this method. Would it be possible for the authors to run this comparison?

I am open to updating my rating if the authors can provide answers to these questions / make edits to the paper that address these concerns.


**Limitations:**

I think the limitations section in the paper is very limited. The only limitation that the authors mention is the difficulty of collecting useful prior dataset, but I think the work has other limitation as well (mentioned in the weaknesses and questions section above) that would be worthwhile to discuss in more detail.

**Strengths And Weaknesses:**

Strengths

- The paper tackles an important problem – how can we compose a diverse set of previously learned skills to solve new tasks quickly?
- The paper shows strong performance when compared to the mostly closely related work - SPiRL.

Weaknesses

- The proposed method has a lot of moving parts: skill encoder / decoders, skill priors (one per skill), downstream policy, Q-function, skill prior generator, adaptive weight module, and so on. All of these various components are closely dependent on each other, and makes me a little concerned about the overall training stability of the method and sensitivity of hyperparameters.
- The experiments are fairly small-scale. The main point of the paper is that it allows combining multiple skill priors to solve the tasks, but two of the experiments involve combining only two skill priors, and one set of experiments involves combining three skill priors. The experiments are also all in navigation domains, so it is unclear if the method will work on other domains, say robot manipulation or Atari games.

---

> ### Author Response · Authors · 2022-08-02
> **Response to Reviewer iPXJ**
>
> We thank the reviewer for all insightful comments. We are encouraged the reviewer can recognize that our work solves an important research problem and find our work has strong performance compared to competitive baselines.
>
>
> > Did you try running the method on domains other than navigation? What did you find? Even negative results can be useful to look at here.
>
> We construct a new robotics manipulation task in robosuite environment. The task is to control a Panda robot arm to grasp the box without colliding with the barrier in the middle of the desk. Our experiments suggest that our approach is able to learn efficiently and converge to high success rates in the robotic manipulation environment. Due to time constraints, we are unable to run all baselines. However, we present the qualitative results (See Fig. 7) to demonstrate ASPiRe’s ability to compose skills. We include the environment details, data collection and experiment results in Additional experiment section A.1. For the reviewer’s convenience, we place the Additional experiment section as the additional appendix in the main paper pdf (after the checklist).
>
>
>
>
>
> > One primary concern was training stability and sensitivity of hyperparameters.
>
> Skill encoder/decoder and all skill priors are trained offline. Those components are freezed during the online training. Given that the training dataset is relatively large, we believe they should not contribute to the instability. Therefore, we focus on the hyperparameters in the online phase.
>
> We have run additional experiments to conduct a hyperparameters sensitivity analysis. We include the experiment details and our analysis in Additional experiment section A.2. For the reviewer’s convenience, we place the Additional experiment section as the additional appendix in the main paper pdf (after the checklist). We summarize the main finding here:
>
>
> There are two hyperparameters that can potentially impact the online learning for the downstream task: a) The target KL divergence $\delta$, which is used to automatically tune the temperature parameters in Eq. 4 and b) The sample size $M$ which is used to estimate the critic value in Eq. 7. We investigate the impact of the sample size $M$ on both Point Maze and Ant Maze environment and the choice of the target KL divergence $\delta$.
>
> We first investigate the impact of sample size M by setting $M=1,10,20$. We find that the sample size has almost no impact on the learning. We hypothesize two reasons: a) The large train batch size in our experiment helps to estimate the critic value $Q(s,\omega)$ more stable and b) Given the input size of weight vector $\omega$ is small, it might be easy for the neural network to approximate its value via small sample size.
>
> We further investigate the impact of the target KL divergence $\delta$. We find that target KL divergence significantly impacts the learning. Based on our experiments, we conclude that a) More complex tasks require higher target KL divergence. The optimal policy to solve complex tasks might be significantly different from the composite skill prior. Therefore, the policy needs more “space” to explore around the composite skill prior. b) Imposing too small target KL divergence can lead to downgraded performance. This is because the policy will be forced to stay close to the composite skill prior and itself might not be able to solve such complex tasks. c) Imposing too big target KL divergence can lead to downgraded performance. As target KL divergence increases, the learned policy will receive less guidance from the prior. Though the learning is sensitive to the choice of target KL divergence, we find that there might still be a range of KL divergences values leading to the same optimal performance.
>
>
>
>
> > Comparison to PARROT
>
> The code for PARROT is not released. Given the time constraints, we are unable to implement PARROT and compare it with our method. The closest baseline compared with PARROT in our evaluation might be BC+Fine tune. In BC+Fine tune framework, we initialize the RL policy with the BC policy learned from offline and use it to explore the environment more efficiently. This is similar to PARROT, which initializes the RL policy with the base distribution used for training the prior. Notice, BC+Fine tune in our evaluation executes actions in the skill space, which is also similar to PARROT which executes actions on z.

---

> > ### Author Response · Authors · 2022-08-08
> > **Response**
> >
> > We’d like to reach out again to check if there were any additional questions or concerns about our rebuttal that we can address before the reviewer-author discussion period ends on August 9. Thanks again for taking the time to read our work and provide helpful feedback!

---

### Official Review · Reviewer_pJKn · 2022-07-11

**Rating:** 6
**Confidence:** 4
**Soundness:** 4 excellent
**Presentation:** 3 good
**Contribution:** 2 fair

**Summary:**

The paper tackles the problem of composing skills from multiple offline datasets. Aspire learns multiple priors from different offline datasets and then learns a set of weights over the priors and a composite prior predictor during online training on new tasks that require application of multiple skills. The method outperforms BC, Multiplicative Compositional Policies (MCP), and SPiRL on a set of 3 Maze tasks that require the agent to compose pushing, avoiding, and navigating actions.

**Questions:**

- Is $G_\mu$ learned offline and $\omega_\sigma$ learned online?
- What $\omega$ is $G_\mu$ conditioned on in equation 6? Should it be $\omega_i$?
- Are all of the baselines trained with the same number of parameters?
- Lines 305-307 state that the push agent fails to complete the setting requiring traversing the maze. But in figure 4 the weight of the push prior is highest and sometimes 1 while traversing the maze. What accounts for this counterintuitive behavior? Why is the push prior so much stronger than navigation in figure 6?

**Limitations:**

The authors describe that their algorithm relies on the presence of labeled skill data for offline training and depends on a set of specified skills in advance.

**Strengths And Weaknesses:**

- Strengths
  - Clarity: The abstract and introduction state the motivation and setup clearly. I found the method section up to 3.3 to be straightforward and concise.
  - Significance: The authors demonstrate that aspire can effectively learn composite policies from offline data of diverse and distinct skills. These results hold even in the presence of distractor skills. Work in this direction enables datasets to be leveraged for a broader set of target control tasks.
  - Quality: The evaluation is thorough: the authors test multiple relevant baselines (MCP, spirl) and show the effectiveness of the method in the presence of irrelevant skills.
  - Originality: To my knowledge, the algorithm and associated losses for a policy regularized by composed primitives is novel.
- Weaknesses
  - Clarity: The explanation of the approach could be made more straightforward.
    - It would be useful to clarify what is learned offline vs online in section 3, especially 3.3.
    - The motivation behind the skill prior generator is difficult to understand because there is no reference to the skill prior generator in the introduction text of section 3 or in figure 2. Based on the previous sections, I expected the adaptive weight module to be a convex combination of the individually learned priors.
  - Quality:
    - A baseline that composes the learned priors with a mixture of experts model (i.e., $G = \sum_{i=1}^Kp_a^i(z|s)$) would clarify the need for the skill prior generator.
    - Figure 6 shows that the push prior accounts for much of the success of the model and Figure 4 shows that, even for maze traversal, the push prior is contributes the most weight to the policy. This counterintuitive result is not explained.
  - Significance:
    - The baselines and prior work present results on more challenging control tasks. E.g., MCP studies tasks on biped, humanoid, and t-rex in addition to ant. SPiRL studies FrankaKitchen.
  - Originality: Section 3.2 could make reference to SVGD [1] and SVPG [2]. SVPG has a similar prior regularization, but for policy learning directly as opposed to skill primitives.

[1] https://arxiv.org/abs/1608.04471
[2] https://arxiv.org/abs/1704.02399

---

> ### Author Response · Authors · 2022-08-02
> **Response to Reviewer pJKn**
>
> We thank the reviewer for insightful and positive feedbacks. We are encouraged the reviewer found that our work is novel in the field of policy regularization literature. We are glad the reviewer recognizes our approach can even work with the presence of distractor skill priors.
>
> > The baselines and prior work present results on more challenging control tasks.
>
> We construct a new robotics manipulation task in robosuite environment. The task is to control a Panda robot arm to grasp the box without colliding with the barrier in the middle of the desk. Our experiments suggest that our approach is able to learn efficiently and converge to high success rates in the robotic manipulation environment. Due to time constraints, we are unable to run all baselines. However, we present the qualitative results (See Fig. 7) to demonstrate ASPiRe’s ability to compose skills. We include the environment details, data collection and experiment results in Additional experiment section A.1. For reviewer’s convenience, we place the Additional experiment section as the additional appendix in the main paper pdf (after the checklist).
>
>
>
>
>
> > It would be useful to clarify what is learned offline vs online in section 3, especially 3.3.
>
> Both weighting function $\omega$ and skill prior generator $G$ is learned online. We have updated our paper to clarify this.
>
> > The motivation behind the skill prior generator is difficult to understand….
> The adaptive weight module evaluates the expected critic value of skills under the composite skill prior distribution generated by weight $\omega$. Notice that the composite skill prior distribution is generated by weighted KL divergence. We select the weights which can lead to the highest expected critic value to assign among primitives. We have updated the paper to clarify the confusion.
>
>
>
> > A baseline that composes the learned priors with a mixture of experts model (i.e., G=∑i=1Kpai(z|s)) would clarify the need for the skill prior generator.
>
> MCP is the baseline that composes the learned priors with a mixture of experts model. As we pointed out in the paper (L245), MCP composes the same set of learned priors as ASPiRe and executes the composite distribution as the policy. Instead of additive Gaussian, it uses the multiplicative Gaussian. In the original MCP paper, they show that MCP has better performance than additive Gaussian. Therefore, we believe MCP is a competitive baseline to compare with.
>
> To further address the reviewer’s concern. We included the additional experiment that uses composite skill prior as a policy in Additional experiment section A.4. For the reviewer’s convenience, we place the Additional experiment section as the additional appendix in the main paper pdf (after the checklist). We observe that using composite skill prior as a policy results in poor performance (similar performance as MCP), which confirms the importance of using composite skill prior as prior instead of policy
>
>
>
> > Section 3.2 could make reference to SVGD [1] and SVPG [2]. SVPG has a similar prior regularization, but for policy learning directly as opposed to skill primitives.
>
> Thanks for pointing them out. We have added them into our updated paper.
>
> > Is Gμlearned offline and $ω_\sigma$ learned online?
>
> Both skill prior generator $G$ and weighting function $\omega_\sigma$ is learned online. We have updated our paper to clarify this.
>
> > What ω is Gμ conditioned on in equation 6? Should it be ωi?
>
> The skill prior generator G in equation 6 is conditional on $\omega$ as a whole. As G generates a composite skill prior, it needs to know the weight assigned to every skill primitive.
>
> > Are all of the baselines trained with the same number of parameters?
>
> Yes. The policy network $\pi$ and critic networks Q have the same number of parameters across all baselines. The number of parameters and architecture for skill encoder/decoder and skill priors are the same for ASPiRe and SPiRL. We used the same neural network architecture and number of parameters reported in the original SPiRL paper.
>
> > Lines 305-307 state that the push agent fails to complete the setting requiring traversing the maze... Why is the push prior so much stronger than navigation in figure 6?
>
> In the grey phase of figure 4, it is true that push prior contributes more weight than navigation. This is because the push primitive includes the skills allowing the ant to approach the box (in a straight line). If the weights are more biased towards navigation prior, the ant will receive less guidance on how to approach the box. Intuitively, the push primitive provides the ant the guidance on which part of the maze it should explore. The navigation prior guides the ant how to explore in the maze. As the ant will only receive the reward if the box is pushed to the target, the push primitive might be the most important one among the three given primitives. Figure 6 also confirms this conclusion.

---

### Official Review · Reviewer_1oiG · 2022-07-12

**Rating:** 6
**Confidence:** 3
**Soundness:** 3 good
**Presentation:** 2 fair
**Contribution:** 2 fair

**Summary:**

This paper extends the prior work on skill prior-regularized RL by learning a skill prior for each skill. When K skill labels are available from the offline data, we can learn K skill polices using VAE as well as K skill priors, which tell an agent whether each skill is plausible to execute given a state. To intermingle K different skill priors into one composite skill prior for downstream RL, a weighting function of K skill priors is learned to maximize the estimated Q-value when following the composite skill prior. One benefit of the composite skill prior is that it enables simultaneous skill composition, e.g. navigating towards a goal while avoiding an obstacle. The experiments demonstrate that the proposed method can effectively combine up to three skill priors, and tackle downstream tasks by not only exectuing one skill at a time but also executing composite skills.


**Questions:**

- Most papers are cited as arXiv papers, which are conference or journal papers. Please use appropriate citation.

- There are many typos (including notations) and grammatic errors. The writing generally needs some improvement.


**Ethics Review Area:**

["I don’t know"]

**Limitations:**

The limitations and potential negative societal impact are well addressed in the paper.

**Strengths And Weaknesses:**

### Strengths

- Figure 3 and 4 are very intuitive and well explain that how the proposed adaptive weight module learns to regulate the downstream task policy.

- The proposed method of weighting different skill priors is novel and the empirical results show the effectiveness of using multiple skill priors quantitatively and qualitatively.

### Weaknesses

- In L28-32, the claim about the single skill prior not being able to handle composition is not very intuitive. Also, Figure 1 does not seem relevant to this context. If this composition is about a mixture of two skills (simultaneous execution of two skills), it should be more explicilty described.

- "the learned policy" sounds weird in some contexts, e.g., L43, L46.

- The assumption of having task labels for trajectories can limit the applicability of the proposed method. The major benefit of the prior work, SPiRL, was in being able to leverage unstructured data.

- It might be good to show the performace of the uniform weights for multiple skill priors to see the effect of the proposed adaptive weight module. Given that only push skill prior can achieve comparable results with ASPiRe in Figure 6, this simple baseline may work as well.

- Is the MCP baseline identical to using the proposed composite skill prior as a policy? Otherwise, it would be important to include this baseline.

---

> ### Author Response · Authors · 2022-08-02
> **Response to Reviewer 1oiG**
>
> We thank for the reviewer for recognizing our approach is novel and effective. We are encouraged that the reviewer find our system can compose multiple skill priors simultaneously and effectively solve the downstream tasks.
>
> We have updated all the citations, and updated the paper writing to improve clarity.
>
> > In L28-32, the claim about the single skill prior not being able to handle composition is not very intuitive. Also, Figure 1 does not seem relevant…it should be more explicilty described
>
> The agent needs to compose navigation skills and avoid skills concurrently when the obstacle is observed (Fig 1) and only activate navigation skills when it is not.
>
> We have updated our paper to describe our motivation to prefer multiple skill priors to compose skills. We have also updated the figure 1 caption to describe the scenario in figure 1 explicitly is about the simultaneous execution of two skills.
>
> > "the learned policy" sounds weird in some contexts, e.g., L43, L46.
>
> We have updated the paper. Thank you for pointing this out.
>
>
> > It might be good to show the performance of the uniform weights…., this simple baseline may work as well.
>
> We have tested the uniform weight in the Ant Maze domain. Specifically, we experimented with two different settings: a) all skill priors are relevant to the downstream tasks b) not all skill priors are relevant to the downstream tasks. We have put the additional experiment results in Additional experiment section A.4. For the reviewer’s convenience, we place the Additional experiment section as the additional appendix in the main paper pdf (after the checklist). Here, we briefly summarize the result.
>
> The reviewer’s hypothesis is correct in setting a) when all primitives are relevant. We observe that uniform weights also deliver good performance. If we compare it with many other random weights assignments,  uniform weights achieve a near-optimal assignment. Regardless, our algorithm is still able to achieve the best performance among all different weight assignments, showing that the system has enough flexibility to learn optimal policy.
>
> However, in the setting b) not all skill priors are relevant to the downstream tasks.
> Our experiment shows that the adaptive weight assignment significantly outperforms the uniform weight. This suggests that our adaptive weight assignment is robust to the various downstream tasks.
>
>
>
>
> > Is the MCP baseline identical to using the proposed composite skill prior as a policy? Otherwise, it would be important to include this baseline.
>
> They are very similar but slightly different. Both of them try to sample the action from composite distribution directly, but the difference is how they composite the distribution. One uses multiplicative gaussian (MCP), and the other uses weighted KL (ours).
>
> We thank for the suggestions and agree it’s an important comparison to include in the paper. We included the additional experiment that uses “proposed composite skill prior as a policy” in Additional experiment section A.4. We observe that using composite skill prior as a policy results in poor performance (similar performance as MCP), which confirms the importance of using composite skill prior as prior instead of policy
>
> > Most papers are cited as arXiv papers, which are conference or journal papers. Please use appropriate citation.
>
> Thanks for pointing it out. We have updated the citation with conference/journal format.
>
> > There are many typos (including notations) and grammatic errors. The writing generally needs some improvement.
>
> Given the time constraints and the number of experiments we need to run, we are only able to fix a limited number of typos. We will carefully go through our paper before we submit our final version.

---

> > ### Comment · Reviewer_1oiG · 2022-08-04
> > **Thanks for your response**
> >
> > Thank the authors for your response. The rebuttal addresses most of my concerns.
> >
> > I read other reviewers' comments and I do agree with the critics on (1) small-scale (simple) experiments compared to MCP and SPiRL and (2) too many moving parts, and I do not think the authors' responses are sufficient to convince the reviewers. I recommend the authors address these comments better than the current responses.

---

> > > ### Author Response · Authors · 2022-08-04
> > > **Response**
> > >
> > > We are glad that our response addressed most of your concerns.
> > >
> > > > 1) small-scale (simple) experiments compared to MCP and SPiRL
> > >
> > > We agree. We construct a new robotics manipulation task in robosuite environment. The environment details, data collection and experiment results are in additional experiment section A.1,
> > >
> > >
> > > > 2) too many moving parts
> > >
> > > We believe that the number of parts is not a metric to measure whether a model is good/bad or whether it is stable or not. However, to address the reviewer’s concern, we can compare the components between ASPiRe and SPiRL and analyze where the instability might come from, and conduct sensitivity analysis in our additional experiment section A.2.   We also want to highlight that we run 8 seeds for all experiments in the main paper and the performance of our model is stable across seeds  ( Fig. 2).
> > >
> > > The extra components ASPiRe compared to SPiRL are a) multiple skill priors instead of one, b) skill prior generator $G(z|s,\omega)$, c) weighting function $\omega$ and an associated optimization procedure. AWM consists of b) and c). Our current paper writing might cause confusion that AWM is another component. We will update our final paper to avoid this confusion.
> > >
> > > For a), to learn each primitive, we use the same procedure that SPiRL trains the single skill prior. Given sufficient training data, primitives should be as stable as the single primitive in SPiRL.
> > >
> > > To validate component b), we present the qualitative results in different domains (Fig 3,4,7) to demonstrate that our learned weight assignment adapts to the tasks on hand. The weight assignment is highly correlated to the skill prior generator $G$, i.e., it evaluates the expected critic value of skills under the distribution generated by $G$. This suggests that the skill prior generator is robust to the different domains.
> > >
> > >
> > >
> > > To validate component c), the optimization procedure for the weighting function depends on the critics. The critic is the common component between ASPiRe and SPiRL, and its learning is independent of the extra components in ASPiRe. Therefore, the instability might only come from estimating the expected critics value of the composite skill distributions. In this process, the sample size M might cause unstable training. Therefore, we have run the hyperparameters sensitivity analysis and found our method can work well even with M=1. The results are in additional experiment section A.2.

---

### Official Review · Reviewer_pGpR · 2022-07-12

**Rating:** 5
**Confidence:** 4
**Soundness:** 3 good
**Presentation:** 3 good
**Contribution:** 3 good

**Summary:**

This paper presents a hierarchical Reinforcement Learning method, where the high-level controller chooses skills that maximize the reward while staying close to a weighted combination of all skill priors. The skills and skill priors are obtained by fitting one skill prior to each labeled dataset using a VAE. A key contribution of this work is the adaptive skill prior, which learns to decide the weighting among skill priors that leads to the highest critic value. The method is evaluated in 3 simulated environments and is shown to be advantageous in terms of sample efficiency and final model performance.

**Questions:**

- As talked about in appendix A.2.1, the skill encoder and decoder are essentially learned from a single primitive and expected to generalize to other primitives. Would this make the skill latent space z more tailored towards the first task?

**Limitations:**

A major limitation of hierarchical learning with fixed primitives is that the task could be impossible to complete if the primitives are not comprehensive enough. Also as the authors have discussed in the paper, separate datasets of different primitives are required. It's unclear how this method will perform in harder downstream tasks that require more primitives.

**Strengths And Weaknesses:**

Strengths:
- The components in the proposed approach are mostly well explained, and the whole idea is sound.
- The Adaptive Weight Module is a novel method to assign optimal weightings among priors to generate skills with high values.
- The visualization of weights along a rollout trajectory is very intuitive.

Weaknesses:
- The experiment environments are from a similar domain. In particular, the point mass and ant differ mainly in terms of low-level dynamics, which mainly challenges the skill learning stage (not the main contribution of this work). Thus, the conclusion drawn seems a bit weak.

---

> ### Author Response · Authors · 2022-08-02
> **Response to Reviewer pGpR**
>
> We first thank the reviewer for the thoughtful and positive feedback. We are encouraged that the reviewer can find the adaptive skill prior to be a contribution to the existing hierarchical Reinforcement Learning literature.
>
> > The experiment environments are from a similar domain… Thus, the conclusion drawn seems a bit weak.
>
> We construct a new robotics manipulation task in robosuite environment. The task is to control a Panda robot arm to grasp the box without colliding with the barrier in the middle of the desk. Our experiments suggest that our approach is able to learn efficiently and converge to high success rates in the robotic manipulation environment. Due to time constraints, we are unable to run all baselines on this new task. However, we present the qualitative results (Fig. 7) to demonstrate ASPiRe’s ability to compose skills. We include the environment details, data collection and experiment results in Additional experiment section A.1. For the reviewer’s convenience, we place the Additional experiment section as the additional appendix in the main paper pdf (after the checklist).
>
>
> > As talked about in appendix A.2.1 … Would this make the skill latent space z more tailored towards the first task?
>
>
> It is incorrect. As discussed in the main paper, section 3.1, the skill encoder and decoder are learned from an aggregated dataset by randomly sampling the state-action tuple from each primitive dataset. The sampling process is uniform among each primitive such that the latent space will not tailor toward any primitive. Later, all primitive skill priors will operate on this shared skill encoder and decoder. Therefore the skill latent space won’t be tailored towards any task in particular. We have added this detail to the appendix to reduce the confusion.
>
>
> > A major limitation of hierarchical learning with fixed primitives is that the task could be impossible to complete if the primitives are not comprehensive enough.
>
> We don’t agree with the assessment. Similar to other prior works (e.g., SPiRL), the policy for the downstream task could benefit from using relevant skill prior (primitives), but not limited to them. For example, if we impose a higher target KL divergence in equation (4). The algorithm will behave like a typical RL algorithm and flexibly learn new skills that are deviating from the skill priors.

---

> > ### Author Response · Authors · 2022-08-08
> > **Response**
> >
> > We’d like to reach out again to check if there were any additional questions or concerns about our rebuttal that we can address before the reviewer-author discussion period ends on August 9. Thanks again for taking the time to read our work and provide helpful feedback!

---

> > ### Comment · Reviewer_pGpR · 2022-08-08
> > **Thanks for your response and clarification!**
> >
> > 1. Result from robotic manipulation task: Thanks for adding this experiment! It definitely adds to the overall soundness of the paper. I strongly encourage the authors to finish some baseline evaluations and add its corresponding plot to Figure 2.
> >
> > 2. Thanks for the clarification! This makes sense.
> >
> > 3. I appreciate the authors' discussion on hierarchical RL and the coverage of skill primitives! Please rest assure that this limitation is not where I base my evaluation because these are open challenges for the entire community.

---

### Author Response · Authors · 2022-08-02
**Summary of response**

We thank the reviewers for their thoughtful feedback. We are encouraged they found our work to be novel (R1, R2, R3), well explained (R1,R2), and our evaluation intuitive (R1,) and thorough (R3). We are pleased they found our approach archives significant improvement (R2, R3, R4) against appropriate baselines (R3, R4). We are glad R4 recognizes that our work tackles an important problem and R3 recognizes that our work enables datasets to be leveraged for a broader set of target control tasks.

One primary concern was the lack of evaluation in the more complex domain (R1,R2,R4), e.g., robotic manipulation. We construct a new robotics manipulation task in robosuite environment. The task is to control a Panda robot arm to grasp the box without colliding with the barrier in the middle of the desk. Our experiments suggest that our approach is able to learn efficiently and converge to high success rates in the robotic manipulation environment. We include the environment details, data collection and experiment results in additional experiment section A.1, Figure 7 and 8.

The other concern was around training stability (R4). We conduct a hyperparameters sensitivity analysis to address the concern. Our experiment shows that our approach is stable with a reasonable choice of parameters. We further conclude the logic to select the parameters and the cause of downgraded performance with an inappropriate choice of parameters. We include the experiment details and our analysis in Additional experiment section A.2 Figure 9 and 10.

R2 and R3 suggest to compare with directly executing composite skill prior as a policy. We conduct and inlcude the comparative experiment in paper additional experiment section.  We observe using composite skill prior as a policy resulting poor performance (similar performance as MCP),  confirming the importance of using composite skill prior as prior instead of policy. We include the experiment details and our analysis in Additional experiment section A.3 Figure 11.

R3 suggests to show the performance of the uniform weights for multiple skill priors to see the effect of the proposed adaptive weight module. We thank for the suggestions and agree it’s an important ablation study to include in the paper. We conduct the experiment and find that our system is able to achieve the best performance among all different weight assignments when all primitives are relevant. However, in the case of irrelevant primitives presented, our experiment shows that the adaptive weight assignment significantly outperforms the uniform weight. This suggests that our adaptive weight assignment is robust to the various downstream task. We include the experiment details and our analysis in Additional experiment section A.4.

We also updated our paper to reduce the confusions that the reviewers had, e.g.,  the training details of skill encoder/decoder (R1,R3), the reasoning for using multiple skill priors (R2) and the motivation behind AWM (R3). We further correct the citation format (R2) and add the missing citation (R3).

We address reviewer comments below in detail and will incorporate all feedback.

---

### Meta-Review · Area_Chair_E1QT · 2022-08-30

**Recommendation:** Accept
**Confidence:** Less certain

**Metareview:**

This paper proposes a method to adaptively combine and use skill priors for reinforcement learning. The setting is very practical and the method proposed is novel and effective empirically. The main concerns from the reviewers were around (1) the experimental setup being too narrow and simple, and (2) the clarity of the paper. The authors added in an additional robotics experiment which partly addressed (1) and provided clarifications in their response to address (2). Overall, this seems like a solid contribution idea-wise that will inspire more work in this direction. I highly encourage the authors to revise the paper in terms of clarity (based on their rebuttal responses) as well as to consider adding a comparison to PARROT (as suggested by reviewer iPXJ).

**Award:**

No

---

### Decision · Program_Chairs · 2022-09-14

Accept